# A Scenario-Based Spatial Multi-Criteria Decision-Making System for Urban Environment Quality Assessment: Case Study of Tehran

Bahare Moradi [1], Rojin Akbari [2], Seyedeh Reyhaneh Taghavi [3], Farnaz Fardad [4], Abdulsalam Esmailzadeh [5], Mohammad Zia Ahmadi [6], Sina Attarroshan [7], Fatemeh Nickravesh [8], Jamal Jokar Arsanjani [9,*], Mehdi Amirkhani [10] and Igor Martek [11]

1    Department of Urban Planning & Design, Faculty of Art and Architecture, Tarbiat Modares University, Tehran 14115-335, Iran; b_moradi@modares.ac.ir
2    Natural Resources Engineering Department, Isfahan University of Technology, Isfahan 84156-83111, Iran; rojeen.abr@gmail.com
3    Department of Urban Planning & Design, Khavaran Institute of Higher Education, Mashhad 91841-68619, Iran; reyhanehtaghavi@yahoo.com
4    Department of Urban Planning, Faculty of Architecture and Urban Planning, Islamic Azad University Qazvin, Qazvin 34185-1416, Iran; ff.farnaz95@gmail.com
5    Department of Social Planning, Faculty of Social Science, Allameh Tabataba'i University, Tehran 15449-15113, Iran; a.esmailzadeh2000@gmail.com
6    Department of Geography, Dr. Ali Shariati Faculty of Letters and Humanities, Ferdowsi University of Mashhad, Mashhad 917794883, Iran; mo.zia.ahmadi@gmail.com
7    Environment Department, Ahvaz Branch, Islamic Azad University, Ahvaz 61349-37333, Iran; sina_2934@yahoo.com
8    Department of Human Geography, Faculty of Geography, University of Tehran, Tehran 14178-53933, Iran; nickravesh.f@ut.ac.ir
9    Geoinformatics Research Group, Department of Planning and Development, Aalborg University Copenhagen, A.C. Meyers Vænge 15, DK-2450 Copenhagen, Denmark
10   UniSA Online, Science Technology Engineering and Mathematics (STEM), University of South Australia, Adelaide, SA 5000, Australia; m.amirkhani@deakin.edu.au
11   School of Architecture and Built Environment, Deakin University, Waterfront Campus, Geelong, VI 3220, Australia; igor.martek@deakin.edu.au
*    Correspondence: jja@plan.aau.dk

**Abstract:** Spatial evaluation of urban environment quality (UEQ) is a key prerequisite in urban planning and development. The main goal of this study is to present a scenario-based spatial multi-criteria decision-making system for evaluating UEQ. Therefore, stakeholder involvement was conducted and eight environmental criteria and six spatial-functional criteria were identified for five districts of Tehran. The weight of the effective criteria was calculated using the analytic hierarchy process (AHP) model. Then, the ordered weighted averaging (OWA) model was used to prepare UEQ maps in different scenarios, including very pessimistic, pessimistic, intermediate, optimistic, and very optimistic. Finally, the spatial distribution of the district population in different classes of UEQ was evaluated. Among the spatial-functional and environmental criteria, the sub-criteria of population density and air pollution, respectively, had the greatest impact on UEQ. In very pessimistic, intermediate, optimistic, and very optimistic scenarios, approximately 76.7, 51.8, 36.4, 23.7, and 9.8 km$^2$ of the studied area had unsuitable UEQ conditions, respectively. In the very pessimistic scenario, about 37,000 and 1,500,000 people lived in areas with suitable and unsuitable UEQ conditions, respectively. In the very optimistic scenario, the population increased to over 917,000 in areas with suitable UEQ and decreased to 336,000 in those with unsuitable UEQ conditions in terms of both environmental and spatial-functional criteria. The research results showed that a high percentage of the population in the study area live under unsuitable UEQ conditions, which indicates the need for attention to improving the current UEQ conditions. The proposed approach is timely to gain a better understanding of the adverse impact of climate change on human well-being in marginal societies and how climate-resilient urban planning can play a significant role.

**Keywords:** urban environment quality; environmental and spatial-functional criteria; spatial multi-criteria decision-making system; geographic information system

## 1. Introduction

In recent decades, population growth and rapid urbanization without proper planning in many developing countries have led to the formation of low-quality urban environments [1–3]. At the same time, governments are under increasing pressure to meet sustainable development goals. This pressure comes not just from the United Nations, but from peer nations who see climate change, environmental degradation, resource depletion, and social inequality as an existential threat that must be met through international collaboration [1,4–8]. On top of that, society at large is increasingly aware of the need to become more sustainable and expects government to deliver more sustainable communities [2,9]. The majority of nations have opted to sign on to the Paris Accord, willingly undertaking to limit carbon emissions to agreed targets by certain fixed dates, and ultimately pledging to become carbon neutral. This is a major challenge, however, and one that many nations are finding hard to attain. While promises to achieve broad, ambitious, macro-level sustainability outcomes might be applauded, they cannot be realized without shifts in how societies are organized and managed. In this regard, urban planning—or the lack of it—has become an existential issue that requires immediate and serious attention [10,11]. Addressing the existential challenges posed by climate change and achieving sustainable development goals requires collaborative efforts at multiple levels and across borders. Governments, urban planners, architects, engineers, community stakeholders, and citizens need to work together to create co-designed and sustainable urban environments that prioritize both human well-being and the health of the planet.

The negative impacts of buildings and infrastructure on the urban environment are well documented [12]. Cities consume up to 80% of global natural resources, produce 75% of greenhouse gas emissions, and generate 50% of all waste [13]. Moreover, where once most people lived in the rural countryside, two-thirds of all people now live in cities. There are now over 20 cities worldwide with over 20 million population, and by 2060, it is anticipated that eight cities will reach 40 million [14]. The exacerbation of climate change is intricately linked to these urban challenges. The excessive consumption of resources and the high greenhouse gas emissions from cities contribute significantly to global warming and environmental degradation. This issue has directly affected the quality of life of citizens [15]. Climate change leads to increased pressure on the urban environment quality (UEQ) and is among the factors exacerbating poorly planned and unsustainable development in cities. Under such circumstances and considering the importance of basic human needs and motivations, the concept of UEQ has emerged in the literature of urban history, planning, design, and urban engineering [16,17]. In recognition of this, the spatial evaluation of UEQ has emerged as a key prerequisite in urban planning and development [18].

According to the annual report of the United Nations, the urban population of Iran will rise to over 85% by 2050, while its rural population will decrease by 36%, reaching less than 15% of the total population [19]. Considering the high and positive annual rate of immigration to major cities of Iran [20,21], and given the natural growth of the population, population growth is expected in the metropolises, especially in Tehran [22]. Forecasts based on the current population trends estimate that by 2030, the Tehran metropolis will experience the phenomenon of urban crowding for the first time in Iran [19]. The population of Tehran has increased by 543 times since it emerged as the capital, and according to the latest census, it has reached more than 8.5 million people, making it one of the largest metropolises in the world [23]. Its area has grown by 139 times since then, reaching 730 km$^2$ [24]. Previous evaluations of the quality of urban life show that the expansion of urban health justice has not been proportional to the growth of the population and area of this city. The existence of inequality within the 22 districts of Tehran has been a

persistent phenomenon, thereby necessitating the need to highlight this inequality [25]. Highlighting these differences through UEQ assessments will foster increased awareness in the organizations and present suitable perspectives on the allocation of financial resources to reduce the UEQ-related class gap. This urban development planning and policy can continue to affect the quality of life of citizens and the development of urban justice. Successful planning requires more assessment and sufficient knowledge of the current state of UEQ.

The purpose of this study is to provide a scenario-based spatial multi-criteria decision-making system for evaluating UEQ. This model was implemented to evaluate the spatial UEQ in five major districts of Tehran using a risk-based expert system based on GIS. To do so, two research questions will be answered: (1) Which of the environmental and infrastructure criteria have a higher impact on the spatial changes of UEQ? (2) How do the area of the UEQ classes and the spatial distribution of the population within them change with the change of the attitude of the decision makers from very pessimistic to very optimistic in the form of different scenarios in the proposed system?

## 2. Literature Review

It is important to model the spatial changes in UEQ and to know the effective criteria for providing solutions to improve the existing conditions [26]. UEQ is a multidimensional concept encompassing environmental and spatial-functional dimensions among others [27]. On the other hand, it is a spatial phenomenon, and spatial and multi-criteria evaluations are necessary to evaluate it [28]. Therefore, accurate modeling of UEQ requires accounting for the impact of different spatial criteria related to environmental and spatial-functional conditions. How the effective criteria affect and become affected by UEQ is complex. Hence, it is vital to employ suitable models for the spatial analysis of UEQ [29,30].

In previous studies, various methods have been used to evaluate UEQ, including questionnaires and data analysis in statistical and spatial software [31–33]. Its disadvantages include underestimating the importance of the location dimension [34], poor quality of the collected data in some cases [35], improper data distribution at the regional level [36], limitations in data collection [37], the complexity of the input parameters [38], misunderstanding of questionnaire items [39], and small statistical population size [34]. Other methods that have been widely used in recent years are tools based on geographic information systems (GIS). The first step in monitoring UEQ is the selection of dimensions and indicators as well as the investigation of the location component in places with dedicated programs and policies [40,41]. GIS provides maps and multivariate statistical analysis to make it possible to examine complex spatial relationships while presenting them to discussion tables for collaborative decision-making [42,43]. Using GIS and spatial display of various urban environment issues can expedite the decision-making process and demonstrate spatial inequalities [44–46]. Also, multi-criteria decision-making (MCDM) approaches can be considered due to their ability to make decisions in situations with multiple and sometimes contradictory criteria [47,48]. MCDM provides a set of techniques and algorithms for structuring the decision problem, while also prioritizing and evaluating the decision problem [49,50].

Many studies have combined MCDM and GIS to generate UEQ maps. Joseph et al. [29] used an expert-based decision-making method combined with GIS to assess UEQ in Port-au-Prince, Haiti. Sadler et al. [51] developed a model based on AHP and GIS to prepare a UEQ map in the city of Flint, Michigan, using 29 environmental and infrastructural indicators. Abd El Karim and Awawdeh [52] investigated UEQ in Buraidah, Saudi Arabia, based on the integration of access and location-allocation models in the GIS environment with a hierarchical decision-making model. Carpentieri et al. [53] investigated UEQ in Naples, Italy, using accessibility and health indicators and a hybrid approach based on GIS and network analysis. Roy et al. [26] presented an approach to assess UEQ using a GIS-based spatial autocorrelation model in India. In this study, to evaluate the quality of

the urban environment, they used 15 indicators in three environmental, human-made, and socio-economic dimensions.

In previous studies, various models based on GIS-MCDA have been used to evaluate UEQ conditions. The results of these models cannot be generalized for different conditions and scenarios, considering that decision-making and planning in the urban environment can be different according to time and budget and the attitude of managers and planners. The OWA system has a high ability to analyze information and provide output for different scenarios. The advantage of this system is flexibility in creating a wide range of different scenarios (very optimistic to very pessimistic), whose results can benefit managers and planners with different perspectives. OWA has been used in various modeling in the fields of renewable energy [42,47], urban growth forecasting [54,55], land suitability [56,57], vulnerability and resilience [44,58], crisis management [59,60], etc. To the best of our knowledge, this study is the first to use this model to assess UEQ conditions.

## 3. Materials and Methods

### 3.1. Study Area

Tehran—the capital of Iran, the largest city in West Asia, and the 19th largest city in the world—is located on the southern slopes of the Alborz mountain range in northern Iran (Figure 1). The daily arrival of people, traveling in and out of Tehran, increases its daily population to more than 20 million people. Tehran has 22 districts, of which Districts 2, 3, 6, 7, and 11 were investigated here. District 2 of Tehran, with a population of about 700,000 people, extends from the north of Tehran and the slopes of Alborz to Azadi Square. The expansion of Tehran over the last three decades from the west has enclosed this area in the north and center of Tehran. Its most notable structures are Milad Tower, Pardisan Forest Park, and numerous hospitals. It also provides easy access to all parts of the city. The vast green space inside this area can be a very important factor in creating leisure time. The concentration of administrative, higher education, and research activities has attracted stakeholders in the district. The existence of vast barren lands next to urban and regional transportation lines is one of the other potentials of the district, they have a very effective role in the establishment of commercial, administrative, and service activities. District 3 of Tehran has a population of about 330,000 people. Due to the lower average rate of population increase in the region compared to the corresponding average rate for the whole city of Tehran, the trend of population changes in District 3 is downward, unlike the city of Tehran. It is somewhat dense and compact in terms of construction and street planning. The existence of major commercial and economic centers, government institutions, and offices brings a large number of people to this district every day, which causes heavy traffic [1,61]. District 6 has a pleasant climate and is located in the northern part of Tehran. According to a 2016 census, its population is about 251,000. This district has old neighborhoods and streets and is also the hub of commercial activities. The ease and variety of access to the public transportation network have increased the relative advantage of the demand for providing regional, urban, national, and even transnational uses in this district compared to other regions of Tehran. District 7 is located in the eastern and central parts of the city, hosting a population of around 312,000 people. In terms of the urban context, there is a significant difference between the western and eastern parts of the district. Thirteen percent of the area of this district has military use, most of which includes vast and uncovered lands. Similar to District 6, the western part of District 7 has administrative and commercial uses. According to the 2016 census, the population of District 11 is about 289,000 people. This district has strategic importance due to its political and strategic centers as well as the presence of important economic centers and specialized extra-regional and extra-urban markets. Two million passengers travel through this district every day and often use the roads of this area to reach other urban areas or to do their administrative work.

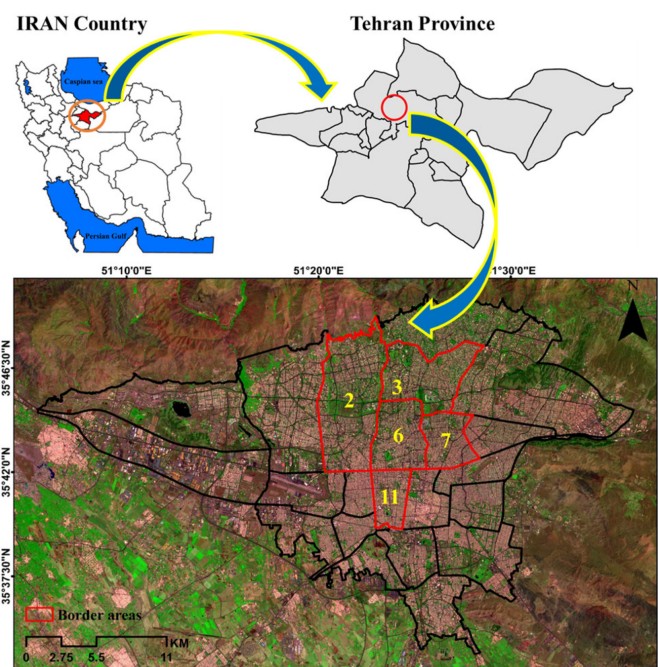

**Figure 1.** Geographical location of the study area (source: authors).

### 3.2. Data Used

Investigation of UEQ and the related indicators requires spatial data. The research data included satellite-based data, location-based field data, and spatial layers collected from related organizations. Satellite-based data included indices of vegetation, discomfort index (DI), albedo, and impervious surfaces coverage (ISC), which were extracted from Landsat 8 imagery for 2022. The spatial resolution of criteria maps obtained from Landsat 8 imagery was 30 m. These satellite images are available at https://earthexplorer.usgs.gov/ (accessed on 2 April 2023). Moreover, the data of the AW3D digital elevation model with a spatial resolution of 30 m were used to prepare the elevation map. These data are available at https://www.eorc.jaxa.jp/ALOS/en/dataset/aw3d30/aw3d30_e.htm (accessed on 5 April 2023). In addition, air pollution indicators were extracted from location-based field data, including fine suspended particles (PM2.5 and PM10), nitrogen dioxide, sulfur dioxide, ozone, and carbon monoxide (https://www.irimo.ir/far/index.php (accessed on 3 April 2023)). Demographic spatial layers were sourced from Iran Statistics Center (https://www.amar.org.ir/ (accessed on 10 April 2023)) and spatial layers of the location of industrial centers, parks, medical centers, subway stations, road networks, river networks, and fault lines were prepared based on maps prepared by Tehran Municipality (https://www.tehran.ir// (accessed on 9 April 2023)).

### 3.3. Methodology

The research process included five steps which are shown in Figure 2. In the first step, previous research, expert opinions, and library sources were used to determine criteria affecting UEQ, and a spatial database was created. Also, this step involved pre-processing operations considering the different sources of the collected data. Then, according to the type of criterion, a map of the criteria was prepared using spatial analysis in GIS. The second step involved standardizing the criteria map. The third step involved calculating criteria weights. In the fourth step, using the ordered weighted averaging (OWA) method, the UEQ map was prepared in different decision-making scenarios for environmental and spatial-functional dimensions. Furthermore, final UEQ maps were prepared by combining these two dimensions. Finally, in the fifth step, an evaluation of population distribution in UEQ classes was conducted.

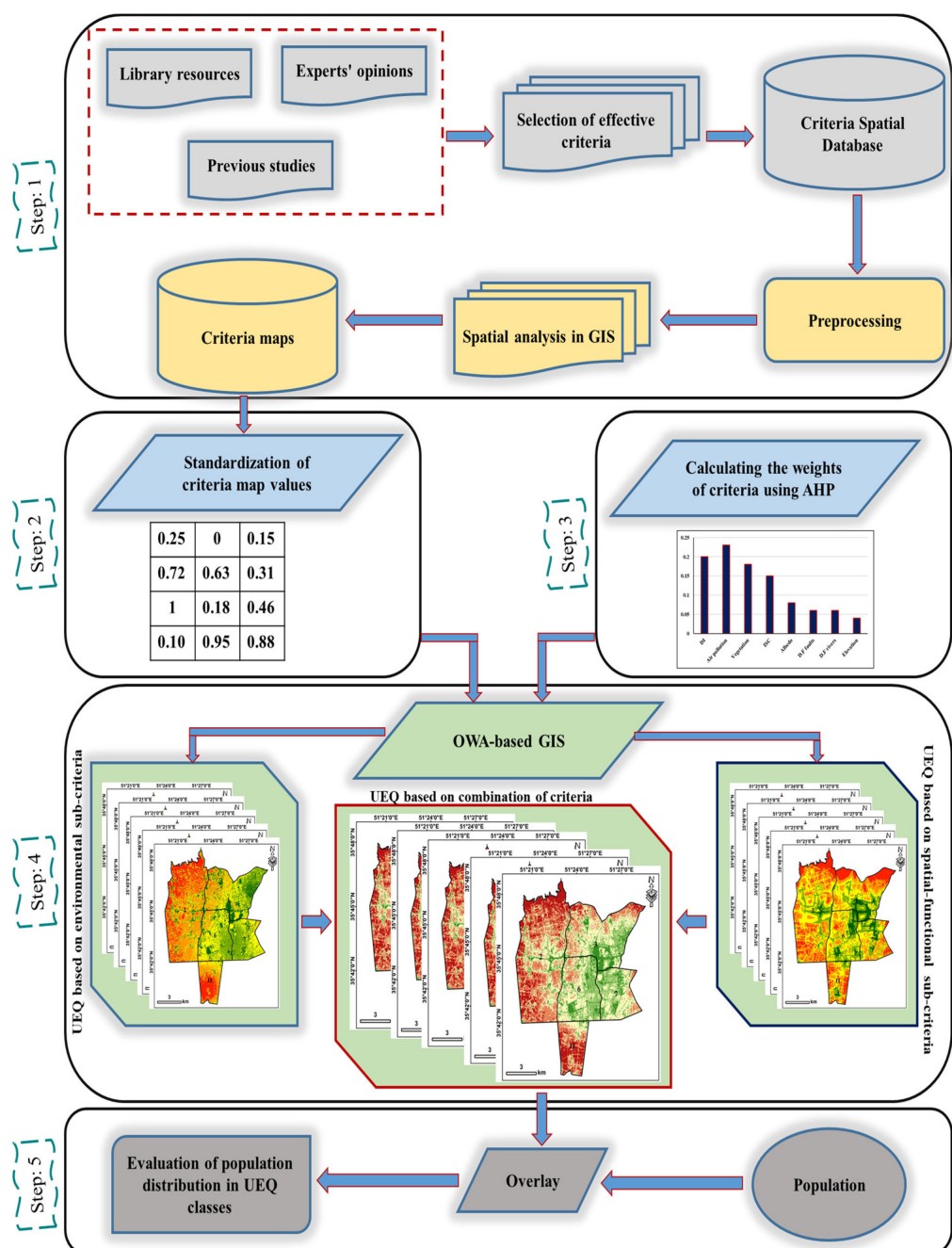

**Figure 2.** The research flowchart (source: authors).

### 3.3.1. Determination of Criteria

Assessing UEQ is an MCDM process that requires careful consideration of a set of criteria. Here these criteria were determined based on previous studies, library sources, expert opinions, and the research conditions. The evaluation criteria were divided into two classes, namely environmental and spatial-functional. The criteria extraction specifications and method are shown in Table 1.

**Table 1.** Descriptions of the criteria used and how to prepare them.

| Criteria | Sub-Criteria | Description | Source |
|---|---|---|---|
| Environmental | DI | DI is an experimental and direct indicator that operates based on the direct measurement of environmental variables [61,62]. This index describes the relationship between the living environment and human thermal sensation [63]. As the value of DI increases, UEQ decreases. In this study, information on bands and spectral indices obtained from Landsat 8 was used to prepare the DI map. The details of the DI calculation method are presented in Mijani et al. [62]. | Landsat 8 and weather station |
| | Air pollution | Air pollution is a major criterion affecting UEQ. Areas with lower air pollution have better conditions in terms of UEQ [64]. This study used data from 15 air pollution measurement stations in Tehran. Also, the inverse distance weighting (IDW) method was used to prepare the air pollution criteria map. | Tehran air quality measurement stations |
| | Vegetation | Vegetation can be effective in modulating the conditions of the urban environment due to its direct effect on air quality, air temperature, relative humidity, pleasant landscape, and creating shade [65]. Areas with a higher degree of vegetation have a higher UEQ. In this study, Normalized Difference Vegetation Index (NDVI) was used to extract vegetation [66]. | Landsat 8 |
| | ISC | ISC can affect UEQ due to its impact on the quality of water resources, thermal pattern control, and landscape. Areas with higher ISC have lower UEQ. Automated built-up extraction index (ABEI) was used to extract ISC [67]. | Landsat 8 |
| | Albedo | The feedback of changes in urban albedo affects the amount of increase or decrease in the outgoing radiation at the earth's surface, leading to climate change [68]. The increase in surface albedo due to changes in urban structure materials has been proposed as a method to adjust urban heat islands or improve energy conservation. In general, light-colored surfaces, such as roof waterproofing surfaces, have a higher albedo than dark surfaces such as asphalt [69]. Vegetation also has a low albedo [70]. WLC method based on Landsat 8 reflective bands has been used to calculate albedo. | Landsat 8 |
| | D.F fault lines and river networks | Proximity to fault lines and river networks can cause serious damage to urban infrastructure in the event of an accident [58]. It can also have financial and deadly consequences for the residents [71]. The Euclidean Distance tool was used to prepare the map of the fault lines and river network. | National Cartographic Center |
| | Elevation | The elevation criterion was used to determine the comfort level of the living environment. This criterion also affects accessibility [72]. Areas located at a higher altitude have rougher topographical conditions and higher slopes, so the access level in these areas is lower. As a result, the quality of the urban environment decreases with the increase in height [73,74]. | Extracted from DEM |
| Spatial-functional | D.F medical centers | People's access to medical facilities (e.g., health centers and hospitals) plays a key role in providing services [75]. Therefore, less distance from medical centers translates into higher UEQ. The Euclidean Distance tool was used to prepare the map of distance from medical centers. | National Cartographic Center |
| | D.F parks | Parks in the urban environment are not only a good place to spend leisure time and have fun. They create a beautiful landscape for residents [76]. Therefore, proximity to parks creates a better UEQ. The Euclidean Distance tool was used to prepare the map of distance from parks. | National Cartographic Center |
| | D.F industrial centers | Industrial centers can increase the amount of urban environmental pollution (e.g., unpleasant odors, smoke, air pollution, and particles), create noise, and increase health risks [77]. Therefore, UEQ increases with distance from these centers. The Euclidean Distance tool was used to prepare the map of distance from industrial centers. | National Cartographic Center |
| | D.F road networks | Adequate road infrastructure in an area can reduce accessibility problems. Proximity to road networks provides residents with easier and faster access to medical centers and other public and service centers [78]. The Euclidean Distance tool was used to prepare the map of distance from road networks. | National Cartographic Center |
| | D.F subway stations | Easy access to public transport stations allows residents to reach their destination faster during traffic jams and road closures [75]. Areas located at a shorter distance from subway stations have a higher UEQ. The Euclidean Distance tool was used to prepare the map of distance from subway stations. | National Cartographic Center |
| | Population density | The high population density in urban areas is a major criterion in evaluating UEQ because it causes pressure on urban resources and creates social conflicts [79]. UEQ decreases with an increase in population density. | National Cartographic Center |

### 3.3.2. Standardization of Criteria

Considering the heterogeneous nature and measurement unit of the research criteria, data standardization is necessary before preparing UEQ maps [80]. By using the standardization method, these heterogeneous values can be converted into 0 and 1. In this

study, "homogenization" and "non-dimensionalization" methods were used to eliminate the dimensions of the criteria. Meanwhile, different criteria affect UEQ differently. For example, distance from industrial centers, distance from river networks, and distance from fault lines have a direct relationship with UEQ. However, DI, air pollution, population density, distance from medical centers, and elevation show an inverse relationship with UEQ. In this study, Equation (1) was used to standardize the criteria having a direct relationship with UEQ, and Equation (2) was used to standardize the criteria having a negative relationship with UEQ.

$$y_{ij} = \frac{Y_{ij} - Y_j^{min}}{Y_j^{max} - Y_j^{min}} \quad (1)$$

$$y_{ij} = \frac{Y_j^{max} - Y_{ij}}{Y_j^{max} - Y_j^{min}} \quad (2)$$

where $y_{ij}$ represents the normalized criterion values, $Y_{ij}$ is the value of the ith position for the jth criterion, and $Y_j^{min}$ and $Y_j^{max}$ are the lowest and highest values of the jth criterion, respectively [81,82].

### 3.3.3. Criteria Weight Calculation

AHP is a popular multi-objective decision-making technique invented by Saaty [83]. This method can be used when the decision-making process is faced with several competing options and decision criteria [54,84]. The final goal of this method is to calculate the weight of criteria and sub-criteria used in a spatial multi-criteria decision-making system. There are four basic steps for using AHP. The first step is to compile the hierarchical structure. A decision hierarchy is a tree that has several levels in relation to the problem under investigation. Its first level expresses the purpose of the decision and its last level expresses the options that are compared with each other and compete for selection. The middle level of this tree is made up of factors that are the criteria for comparing options. The second step involves forming the matrix of pairwise comparisons. In this step, the elements of each level are compared to other related elements at a higher level in a pairwise manner, and matrices of pairwise comparisons are formed. A range of 1 to 9 [83] was used to determine the importance and preference in pairwise comparisons. The third step includes calculating the weight of criteria and sub-criteria. At this stage, arithmetic and geometric averaging methods are used to calculate the weight based on the matrix of pairwise comparisons. The fourth step is to calculate the consistency rate (CR). The compatibility rate is a mechanism that shows the level of trust in the obtained priorities. So that if the CR is less than 0.1, the compatibility of the comparisons can be accepted, otherwise the comparisons must be performed again [85].

### 3.3.4. OWA Method

Multi-criteria evaluation methods in GIS usually include a set of spatial evaluation criteria in the form of maps and layers. But the problem that usually arises in spatial decision-making is how to combine criteria maps with a set of weights and also account for the priorities of decision makers. Yager [86] introduced the OWA operator, which can calculate risk aversion and risk acceptance in individuals and apply it to the final option. So far, various methods have been presented for the MCDM process, including the method of zero and one operators (Boolean) with non-compensatory combination rules [87] and the weighted linear combination (WLC) method with compensatory combination rules [88]. OWA is more general than these methods. This operator, like all decision operators, assumes an n-dimensional space on a one-dimensional space [89]. For example, assuming an MCDM problem concerning the integration of factor maps, the ordinal weights of the pixel values in each map are normalized and ordered, then, the criterion score of each map is the relative importance weight of the criteria. Therefore, the formula for the obtained pixel values is in the form of Equation (3):

$$\text{OWA} = \sum_{j=1}^{n} \left( \frac{u_j v_j}{\sum_{j=1}^{n} u_j v_j} \right) z_{ij} \tag{3}$$

where $v_j$ is the reordered j-th attribute weight according to the reordered attribute value $z_{ij}$, $u_j$ is the weight of j-th criterion for all locations to indicate the relative importance of the attribute according to the decision maker's preferences, and $z_{ij}$ is the value of the $i^{th}$ cell according to the $j^{th}$ criterion [90].

The OWA operator includes two important indicators that express its behavior and position: (i) the degree of ORness and (ii) the amount of tradeoff [91]. The degree of ORness shows the position of the OWA operator between AND and OR relationships and expresses the decision maker's risk aversion and risk-taking behavior [92]. Meanwhile, the tradeoff shows the level of compensability of a measure [93]. This means that one poor criterion weight is compensated for by a high criterion weight on other factors. In the OWA method, the decision strategy space is defined based on two indicators, Tradeoff and ORness (Figure 3). In this space, the horizontal axis represents the amount of risk that extends from a space without risk (AND) to a space with high risk (OR). The vertical axis also represents the compensation between the criteria and extends from the space without compensation to one with high compensation.

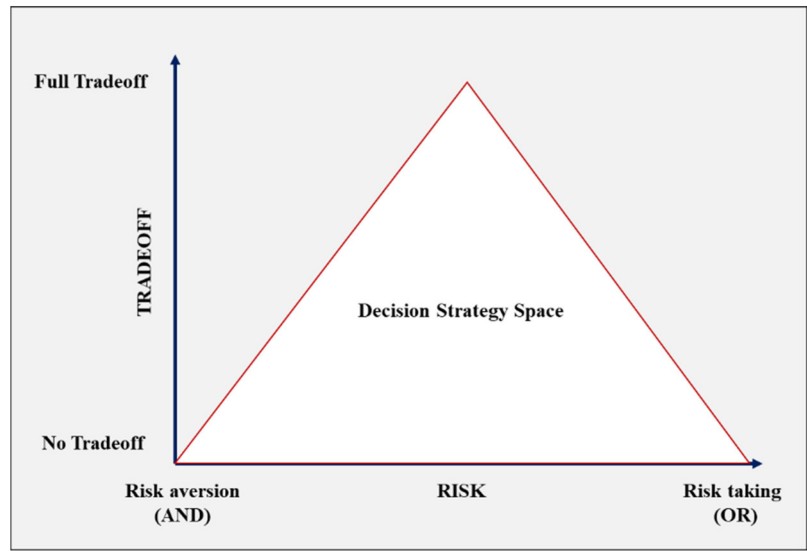

**Figure 3.** Decision-making strategy space in the OWA method [94].

3.3.5. Population Distribution in UEQ Classes

In this study, based on the OWA method, UEQ maps are prepared for the study area in different scenarios, including very optimistic, optimistic, intermediate, pessimistic, and very pessimistic. The UEQ value varies from 0 (low UEQ) to 1 (high UEQ). Also, UEQ maps are classified into five classes based on the UEQ grade: very low (0–0.2), low (0.2–0.4), medium (0.4–0.6), high (0.6–0.8), and very high (0.8–1). Then the area of each class is calculated in different scenarios. Finally, the state of population distribution in different UEQ classes is evaluated.

## 4. Results

### 4.1. Criteria Weight

The weight obtained for each of the effective criteria based on the opinion of experts is shown in Figure 4. The weight of each criterion indicates its degree of importance in the final decision. By changing the weight of a criterion, the degree of importance of that criterion changes in decision-making. In other words, by increasing the weight of each criterion, its importance increases, and conversely, by decreasing its weight, its importance

decreases for UEQ modeling. The weight of the criteria varies between 0 and 1. In this study, in the environmental dimension of the criteria, air pollution, discomfort index, and vegetation had the highest weight, and elevation, distance from river networks, and distance from fault lines had the lowest weight. In terms of infrastructure, the criteria of population density and distance from the road networks had the highest weight, and distance from industrial areas and distance from medical centers had the least weight. The compatibility rate of the determined weights for spatial-functional and environmental criteria was 0.003 and 0.002, respectively, which shows the consistency of the opinions of different experts in determining the weight of these criteria. The weight of spatial-functional and environmental criteria was 0.46 and 0.54, respectively.

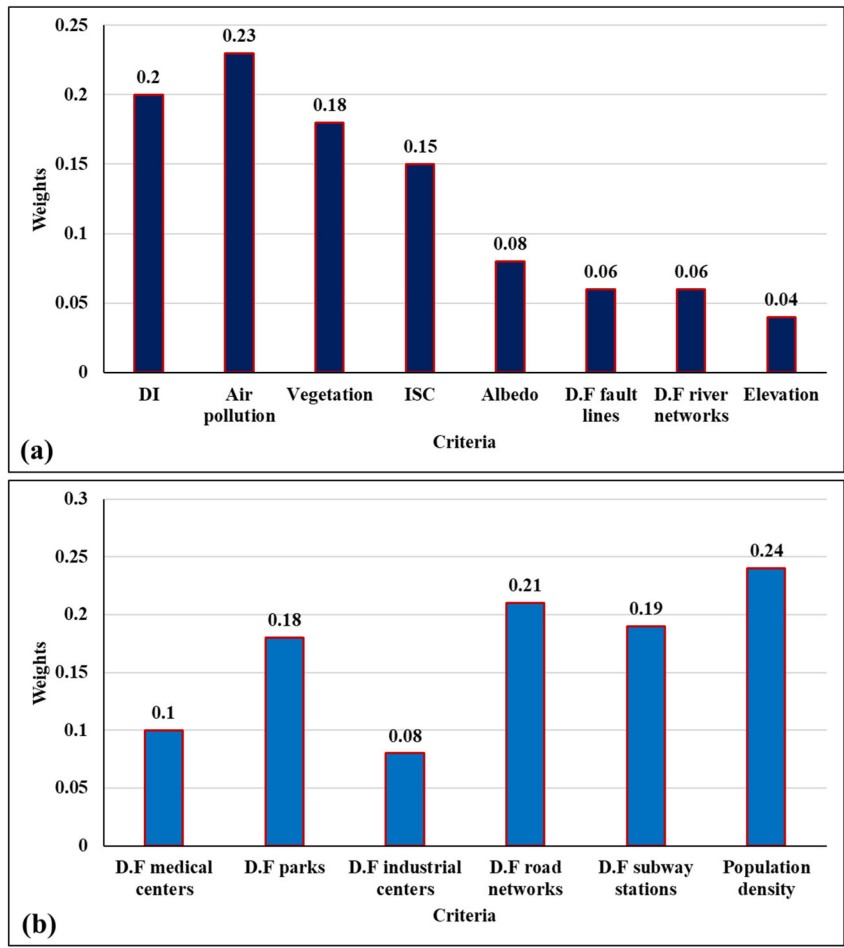

**Figure 4.** The weight of the criteria used; (**a**) environmental criteria and (**b**) spatial-functional criteria (source: authors).

### 4.2. Maps of Environmental and Spatial-Functional Criteria

The spatial distribution of the values of environmental criteria affecting UEQ is shown in Figure 5. Areas with criteria values of 1 (red) had the best UEQ conditions, whereas those with criteria values of 0 (blue) had the worst UEQ conditions. In general, according to the AQI index, air pollution was high in the study area. In terms of spatial changes, air pollution showed a north–south trend, with the highest air pollution in the southern districts and the lowest in the northern districts. In terms of air pollution criteria, the northern and northeastern districts had the best UEQ conditions, while the southern and southwestern districts had the worst UEQ conditions. In terms of discomfort index criteria, barren lands and densely built lands had unsuitable UEQ. These lands are located in District 2. Moreover, parks and green areas had the best thermal comfort conditions, and so the highest UEQ. In terms of spatial variations of vegetation, the northern and central parts

of the study area had better UEQ conditions than the southern parts. This is due to the presence of larger parks and tree-lined streets in the northern and central parts. The spatial variations of the ISC criterion were close to the spatial variations of the vegetation criterion. According to the spatial changes of albedo, the central and northwestern districts had better UEQ conditions than other districts. The southern areas had a lower elevation than the northern areas, which indicates better UEQ conditions in the former. A fault passes through the northwestern part, creating unsuitable UEQ conditions in terms of distance from faults. In terms of distance from river networks, District 11 had a better UEQ condition than the other 4 districts.

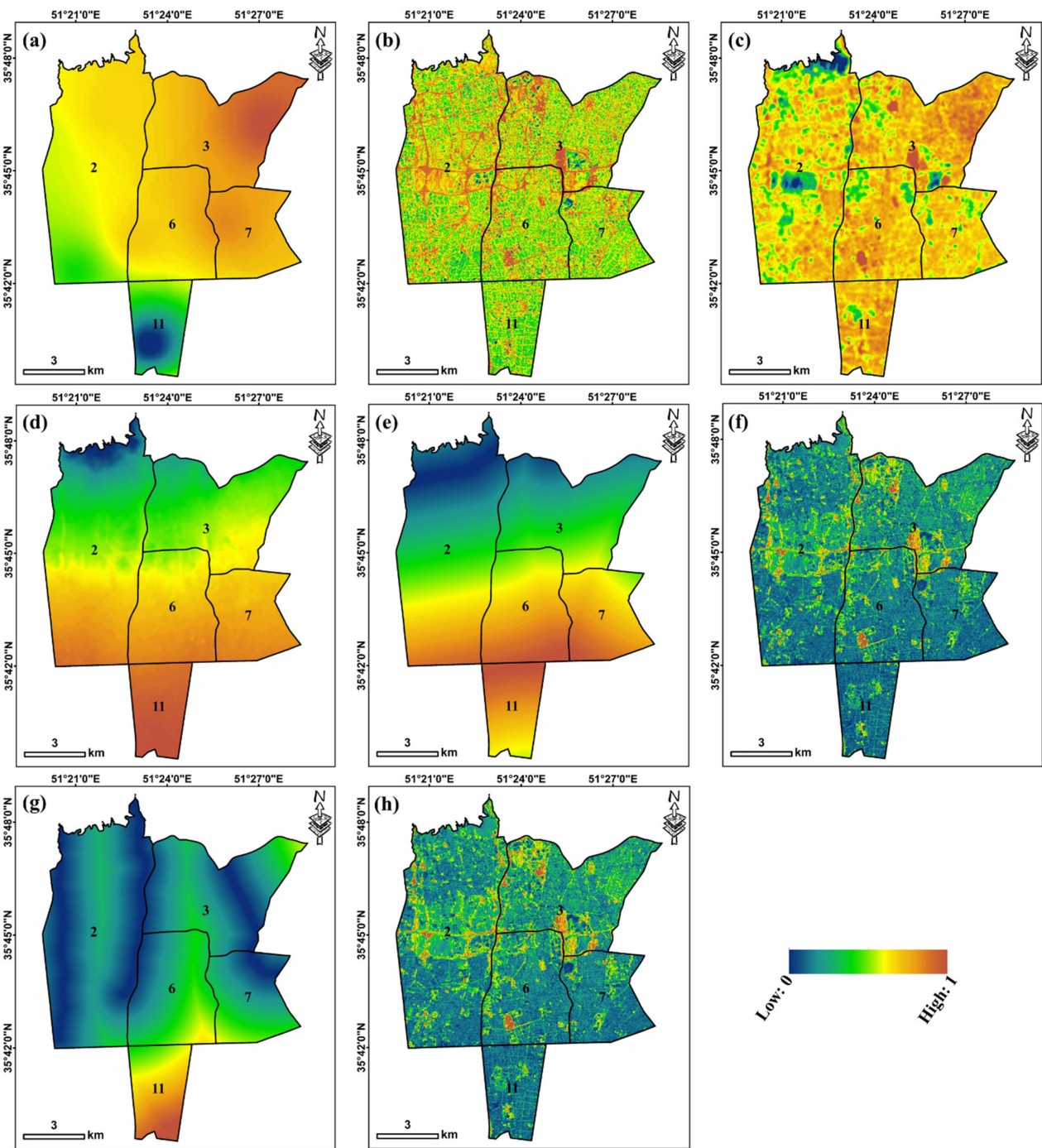

**Figure 5.** Maps of environmental criteria; (**a**) air pollution, (**b**) albedo, (**c**) DI, (**d**) elevation, (**e**) D.F fault lines, (**f**) ISC, (**g**) D.F river networks, and (**h**) vegetation (source: authors).

The spatial distribution of the values of spatial-functional criteria affecting UEQ is shown in Figure 6. Eastern and northwestern regions had a more suitable distance from industrial centers. Therefore, Districts 11 and 6 have more unfavorable UEQ conditions in terms of distance from industrial centers, but better UEQ conditions in terms of distance from medical centers. In terms of distance from medical centers, the northeastern and eastern sectors had the worst condition. Most of the medical centers are located in the southern and central regions. The main reason is the high population density in these areas. In general, the distribution of parks at the district level is relatively suitable and increases from the south to the north. The reason is the existence of more open spaces and fewer residential areas towards the northern regions of the study area. The density of parks was higher in Districts 2 and 6. Districts 3 and 11 had poorer UEQ conditions than other districts in terms of distance from parks. District 11 had the highest population density, while District 3 included more undeveloped lands and parks, which have the lowest population density. Therefore, in terms of population density, Districts 3 and 11 had the best and worst UEQ conditions, respectively. The density of the road network in the southern regions of the study area was higher than in the northern regions. In terms of distance from subway stations, the southern regions had better UEQ conditions than the northern regions.

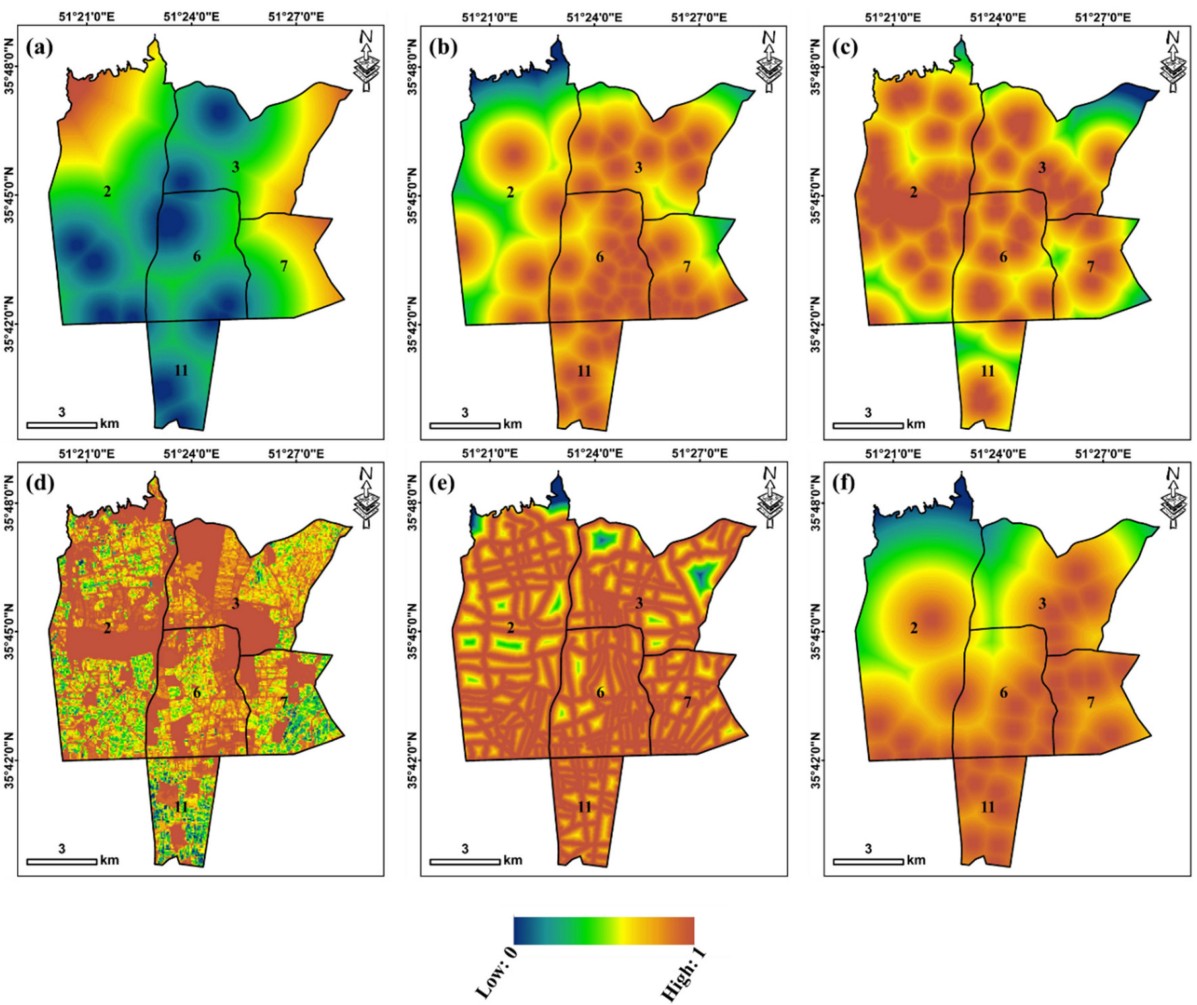

**Figure 6.** Maps of spatial-functional criteria; (**a**) D.F industrial centers, (**b**) D.F medical centers, (**c**) D.F parks, (**d**) population density, (**e**) D.F road networks, and (**f**) D.F subway stations (source: authors).

### 4.3. UEQ Based on Environmental Criteria

The classified maps of UEQ based on environmental criteria for different degrees of ORness are shown in Figure 7. The UEQ value varies from 0 (low UEQ) to 1 (high UEQ). These maps are classified into five classes based on the UEQ grade: very low (0–0.2), low (0.2–0.4), medium (0.4–0.6), high (0.6–0.8), and very high (0.8–1). The visual evaluation of the maps shows that the eastern half of the study area has a more favorable UEQ situation than the western half. This is due to the adjustment of the urban microclimate in accordance with the increase in vegetation, reduction in air pollution, and improvement of DI towards the eastern parts (mountain range), which directly affects other environmental indicators. Districts 1, 6, and 7 had better UEQ conditions than Districts 2 and 11.

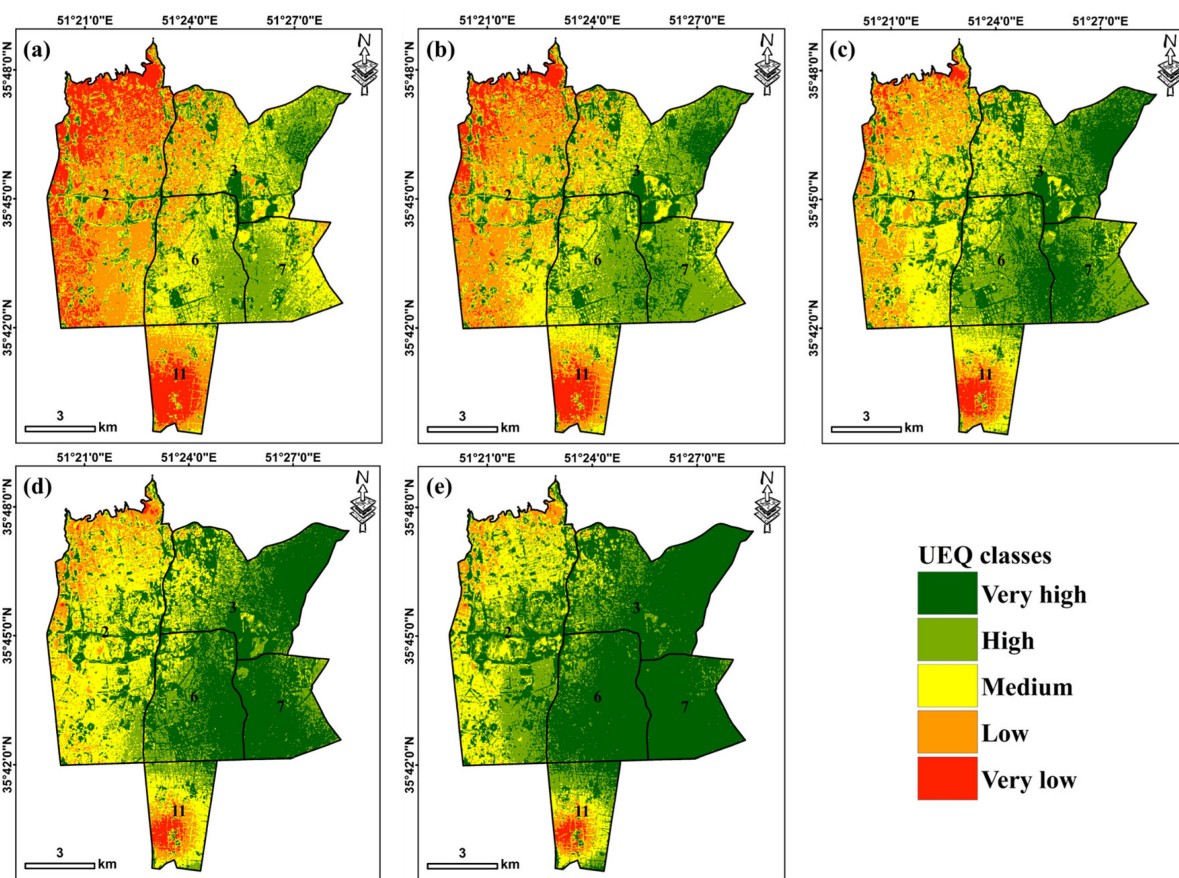

**Figure 7.** UEQ maps based on environmental criteria in different ORness; (**a**) Orness = 0, (**b**) Orness = 0.25, (**c**) Orness = 0.5, (**d**) Orness = 0.75, and (**e**) Orness = 1 (source: authors).

The coverage percentage of different UEQ classes based on environmental criteria for different degrees of ORness is shown in Figure 8. The coverage percentages of the very high UEQ class in terms of environmental criteria for very pessimistic, pessimistic, intermediate, optimistic, and very optimistic scenarios were 9, 16, 30, 43, and 57 percent, respectively. These values for the very low class were 14, 8, 3, 2, and 1 percent, respectively. These results show that by increasing the ORness value or increasing the degree of optimism, the area of the very high UEQ class has significantly increased, while the area of the very low UEQ class has decreased. In terms of environmental criteria, in the very pessimistic scenario, 3.5, 17.6, 10.9, 9.16, 1.0, and—percent of Districts 1, 2, 3, 6, 7, and 11 had very high UEQ conditions, respectively. In the case of a very optimistic scenario, these values for these districts increased to 24.4, 80.3, 90.2, 97.5, and 23.2. In the intermediate scenario, respectively, 10.5, 47.3, 43.3, 53.7, and 4.4 percent of Districts 1, 2, 3, 6, 7, and 11 had very high UEQ conditions in terms of environmental criteria.

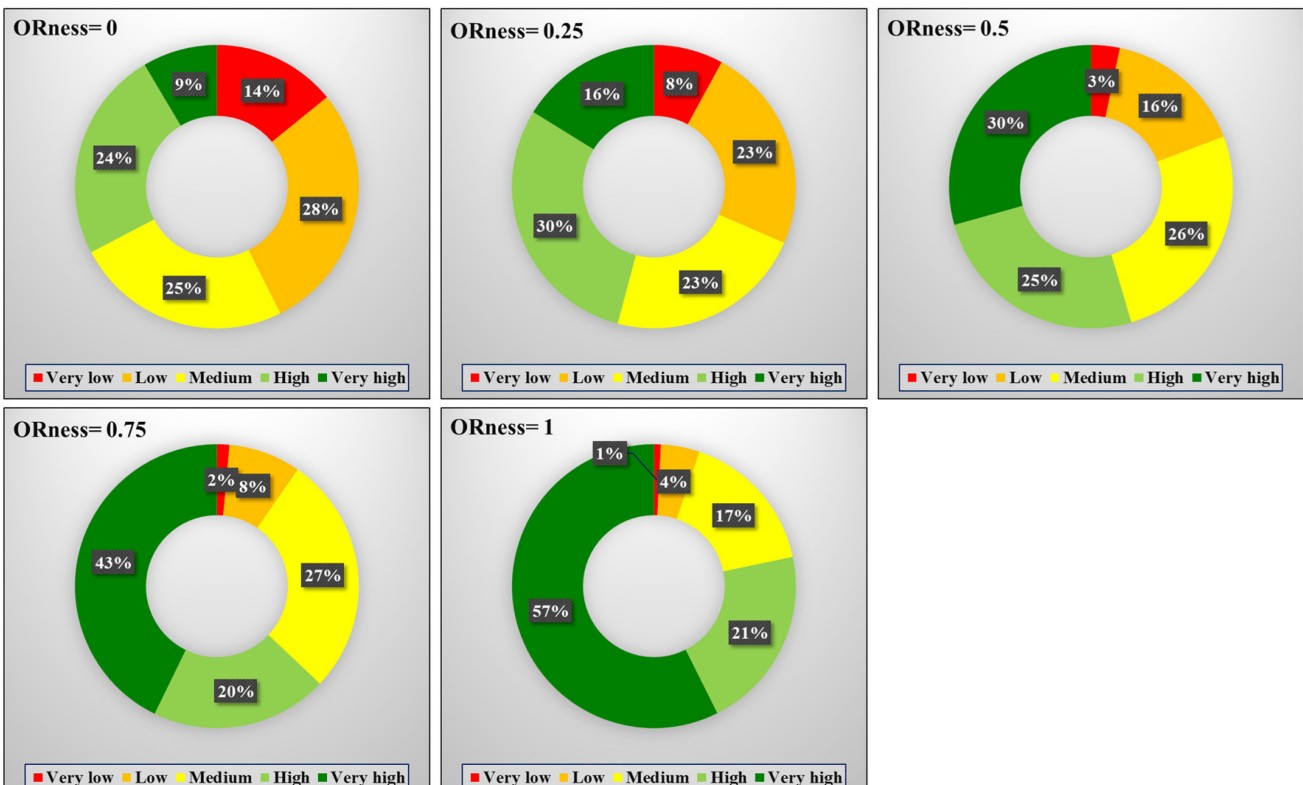

**Figure 8.** Area percentage of different UEQ classes for each ORness in the environmental dimension (source: authors).

### 4.4. UEQ Based on Spatial-Functional Criteria

UEQ classification maps based on spatial-functional criteria for different ORness degrees are shown in Figure 9. The visual evaluation of the maps shows that in terms of criteria related to the spatial-functional dimension, the districts with high and very high UEQ conditions are almost evenly distributed throughout the region. The reason for this is the heterogeneity in the quality of access and distribution of urban services in these districts. In general, the central parts have a more favorable UEQ status than other parts. The infrastructure of the road networks, subway stations, medical centers, and parks is more concentrated in these areas. In terms of spatial-functional criteria, the areas located in the northern parts had the worst UEQ conditions. District 2 did not have suitable UEQ conditions in terms of environmental criteria, while the central parts of this region showed suitable UEQ conditions in terms of spatial-functional criteria.

The coverage percentage of different UEQ classes based on spatial-functional criteria for different degrees of ORness is shown in Figure 10. The coverage percentage of very low UEQ class in terms of spatial-functional criteria for very pessimistic, pessimistic, intermediate, optimistic, and very optimistic scenarios was 15, 7, 4, 3, and 2 percent, respectively. These values for the very high class were 6, 14, 25, 39, and 56 percent, respectively. These results show that the area of the very high UEQ class has significantly decreased and the area of the very low UEQ class has increased with the increase in the ORness value or the increase in pessimism. In terms of spatial-functional criteria, in the very optimistic scenario, 44.2, 54.8, 82.2, 68.3, and 43.4 percent of Districts 2, 3, 6, 7, and 11 had very high UEQ conditions, respectively. In the case of a very pessimistic scenario, these values were reduced to 1.5, 9.0, 11.5, 10.3, and 2.8. In the intermediate scenario, respectively, 15.2, 27.6, 41.9, 35.5, and 15.9 percent of Districts 2, 3, 6, 7, and 11 had very high UEQ conditions in terms of spatial-functional criteria.

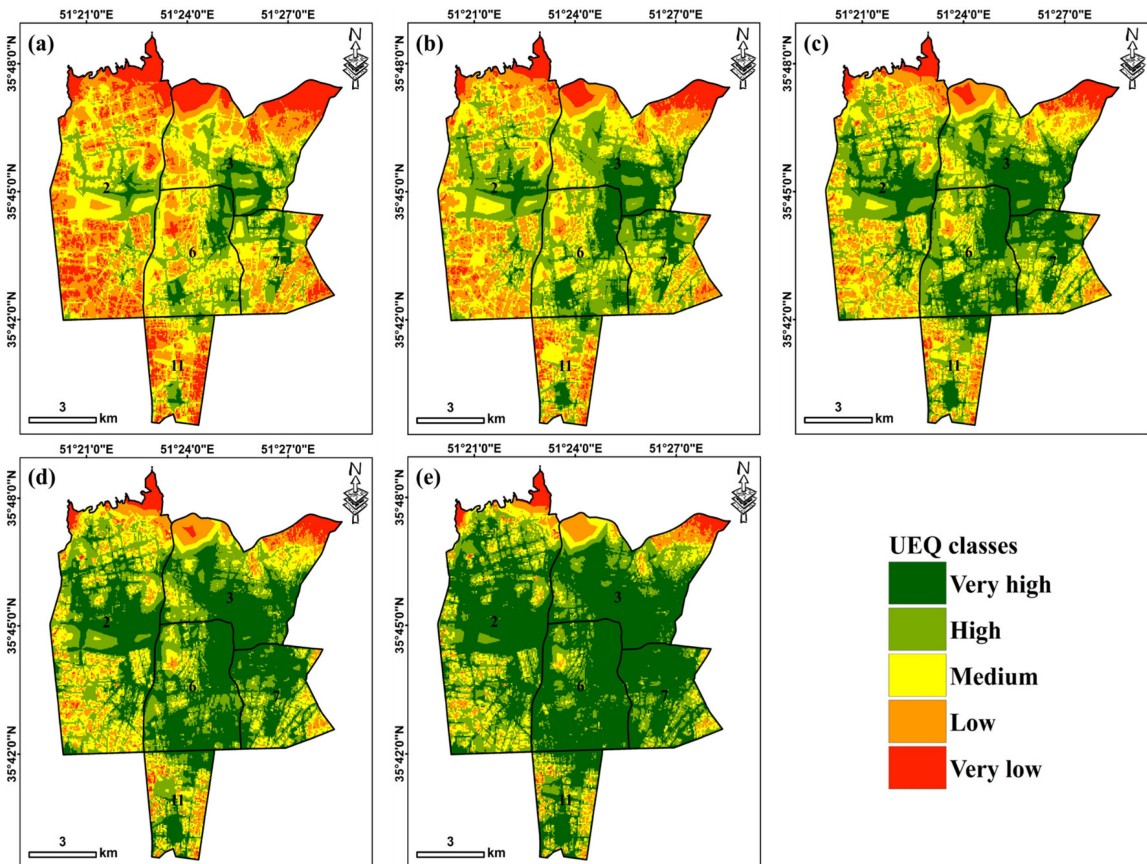

**Figure 9.** UEQ maps based on spatial-functional criteria in different ORness values; (**a**) Orness = 0, (**b**) Orness = 0.25, (**c**) Orness = 0.5, (**d**) Orness = 0.75, and (**e**) Orness = 1 (source: authors).

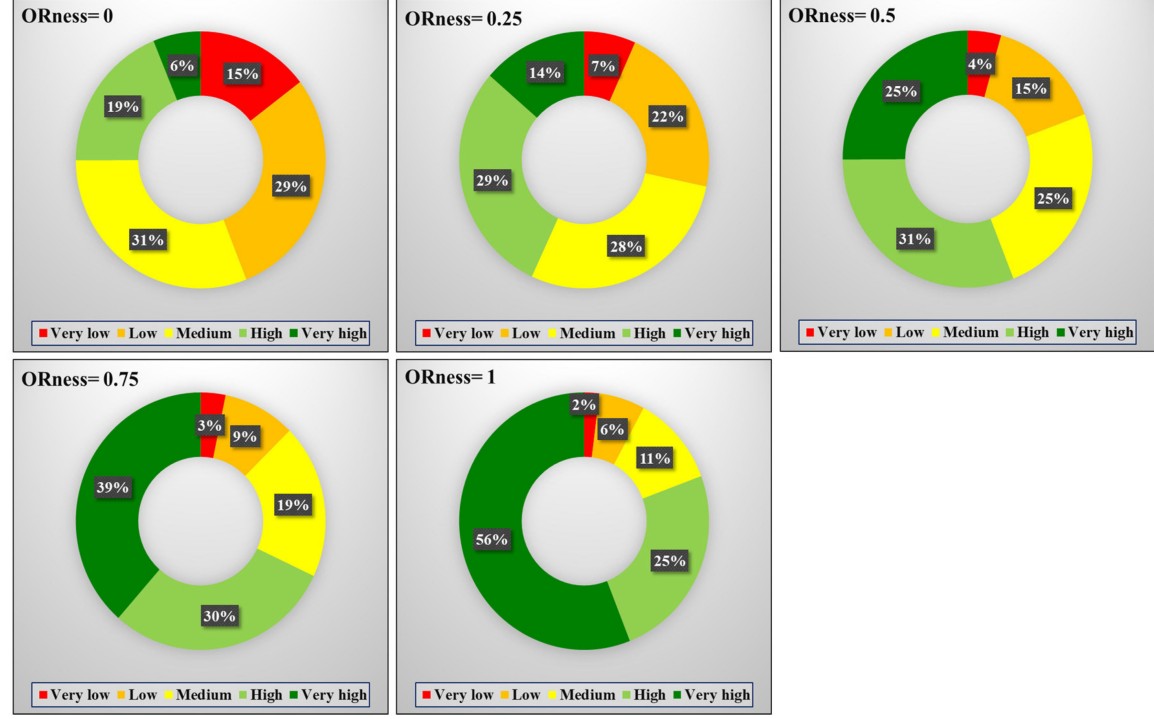

**Figure 10.** Area percentage of different UEQ classes for each ORness in the spatial-functional dimension (source: authors).

### 4.5. UEQ Based on Environmental and Spatial-Functional Criteria

The UEQ maps were prepared by combining the UEQ maps based on environmental and spatial-functional criteria using the WLC model (Figure 11). A visual inspection of these maps shows that the central and eastern regions have better UEQ conditions than other regions. Districts 2 and 11 had worse UEQ conditions than other districts. By increasing the value of ORness, the area of the high and very high UEQ classes increased and the area of the low and very low UEQ classes decreased.

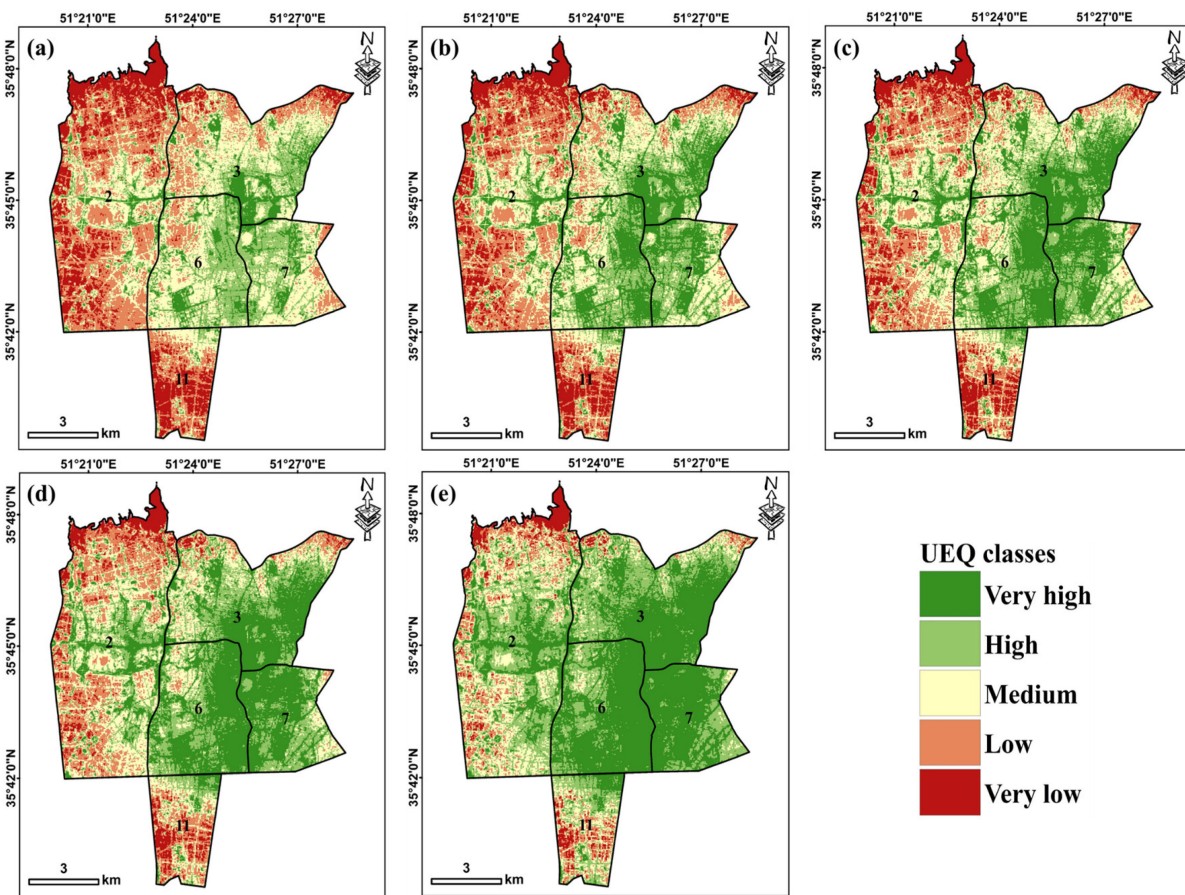

**Figure 11.** The final UEQ maps (combined environmental and spatial-functional dimensions); (**a**) ORness = 0, (**b**) ORness = 0.25, (**c**) ORness = 0.5, (**d**) ORness = 0.75, and (**e**) ORness = 1 (source: authors).

The area of UEQ classes for different degrees of ORness is shown in Figure 12. The area of very low, low, medium, high, and very high UEQ classes in the very pessimistic scenario was 2.17, 33.5, 38.8, 25.4, and 11.2 km$^2$, respectively. These values were 11.2, 24.4, 32.3, 41.0, and 54.3 km$^2$ in the very optimistic scenario. The total area of regions with high and very high UEQ in very pessimistic, pessimistic, intermediate, optimistic, and very optimistic scenarios was approximately 36, 50, 58, 71, and 91 km$^2$, respectively. The total area of regions with low and very low UEQ in these scenarios was 50, 43, 35, 27, and 14 km$^2$, respectively.

In another analysis, by overlapping the UEQ class maps based on environmental and spatial-functional criteria, the state of the study area was evaluated in terms of the agreement between these two groups of criteria in different ORness values (Figure 13). In ORness = 0 (a very pessimistic scenario), most of the study area showed unfavorable UEQ conditions in terms of both groups of environmental and spatial-functional criteria. In all scenarios, the northeastern parts had good UEQ conditions only in terms of environmental criteria. Parts of the western and northwestern regions had suitable UEQ conditions only

in terms of spatial-functional criteria. The areas with suitable UEQ conditions in terms of both environmental and human criteria were located in the central and eastern parts of the study area.

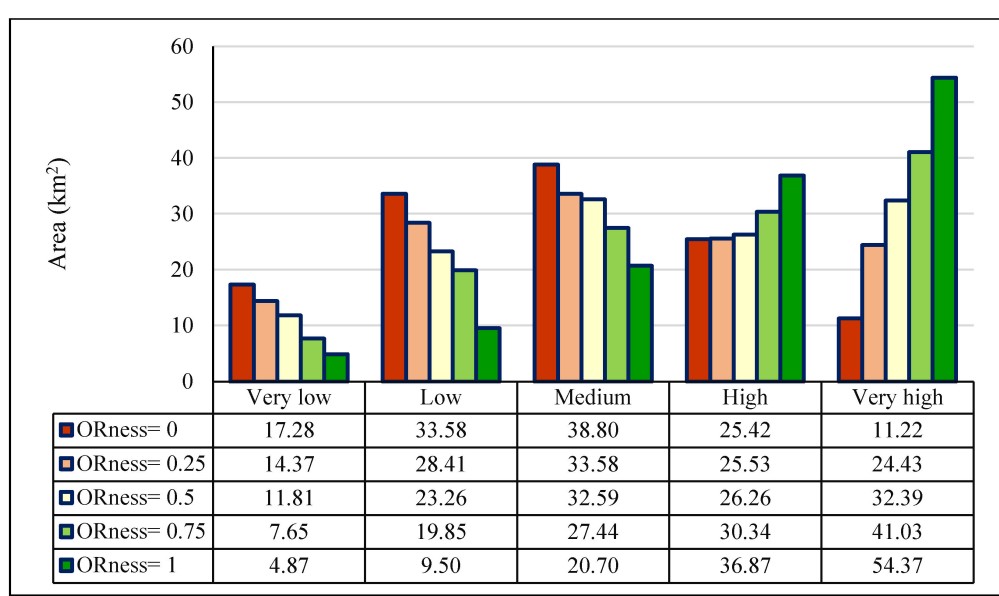

| | Very low | Low | Medium | High | Very high |
|---|---|---|---|---|---|
| ORness= 0 | 17.28 | 33.58 | 38.80 | 25.42 | 11.22 |
| ORness= 0.25 | 14.37 | 28.41 | 33.58 | 25.53 | 24.43 |
| ORness= 0.5 | 11.81 | 23.26 | 32.59 | 26.26 | 32.39 |
| ORness= 0.75 | 7.65 | 19.85 | 27.44 | 30.34 | 41.03 |
| ORness= 1 | 4.87 | 9.50 | 20.70 | 36.87 | 54.37 |

**Figure 12.** The area of different UEQ classes for each ORness in the final UEQ maps (km$^2$) (source: authors).

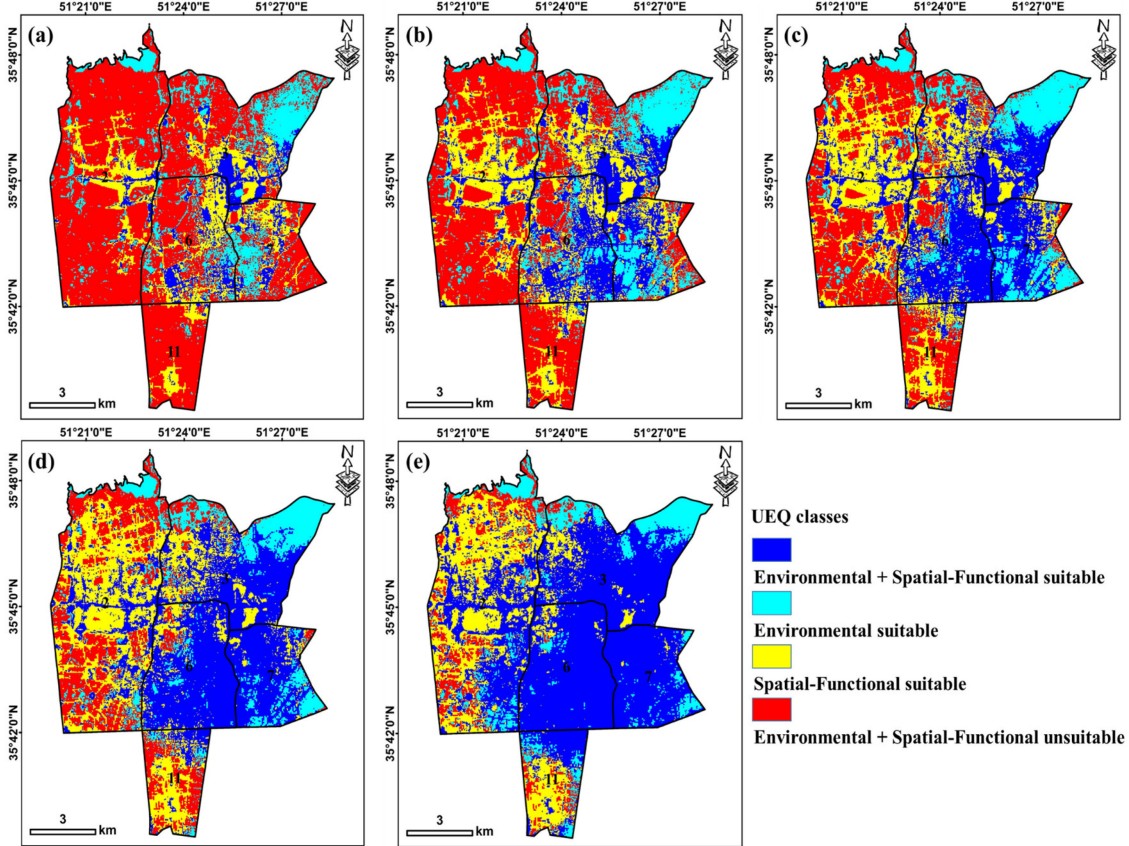

**Figure 13.** Maps of areas with suitable UEQ spatial agreement based on both environmental and spatial-functional criteria in different scenarios including **a**) ORness = 0, (**b**) ORness = 0.25, (**c**) ORness = 0.5, (**d**) ORness = 0.75, and (**e**) ORness = 1 (source: authors).

The area of UEQ classes based on the spatial overlap of suitable UEQ classes obtained from the environmental and spatial-functional criteria groups is shown in Table 2. In the very pessimistic, pessimistic, intermediate, optimistic, and very optimistic scenarios, approximately 11.9, 25.9, 38.8, 51.9, and 74.1 km$^2$ had suitable UEQ conditions in terms of both groups of environmental and spatial-functional criteria. Moreover, in these scenarios, respectively, 17.8, 20.0, 19.1, 16.7, and 14.6 km$^2$ had suitable UEQ conditions only in terms of environmental criteria. These values for the spatial-functional criteria group were 19.8, 28.5, 31.8, 33.8, and 27.8 km$^2$, respectively. In addition, in very pessimistic, pessimistic, intermediate, optimistic, and very optimistic scenarios, approximately 76.76, 51.8, 4.36, 23.7, and 9.8, respectively, had unsuitable UEQ conditions in terms of both criteria groups.

**Table 2.** The area of classes with suitable UEQ spatial agreement based on both environmental and spatial-functional criteria in different scenarios (source: authors).

| | Environmental + Spatial-Functional Suitable | Environmental Suitable | Spatial-Functional Suitable | Environmental + Spatial-Functional Unsuitable |
|---|---|---|---|---|
| ORness = 0 | 11.92 | 17.81 | 19.83 | 76.74 |
| ORness = 0.25 | 25.92 | 20.01 | 28.52 | 51.87 |
| ORness = 0.5 | 38.83 | 19.15 | 31.86 | 36.47 |
| ORness = 0.75 | 51.91 | 16.76 | 33.88 | 23.76 |
| ORness = 1 | 74.10 | 14.60 | 27.81 | 9.80 |

### 4.6. Population in UEQ Classes

The population in different classes of spatial agreement under suitable UEQ conditions for both criteria groups in different scenarios is shown in Table 3. In the pessimistic scenario, only about 37,000 people live in areas with suitable UEQ conditions in terms of both criteria groups, while over 1,500,000 people live in unsuitable UEQ conditions. In the very optimistic scenario, the population distributed in areas with suitable UEQ conditions in terms of both criteria groups increased to more than 917,000 and those in unsuitable UEQ conditions decreased to 336,000 people. In the intermediate state, respectively, more than 349, 453, 212, and 947 thousand people are located in areas with suitable UEQ conditions in terms of "Environmental + spatial-functional", "Environmental" and " spatial-functional", and areas with unsuitable UEQ conditions.

**Table 3.** The population in different classes of the spatial agreement of suitable UEQ conditions based on both criteria groups in different scenarios (Source: authors).

| | Environmental + Spatial-Functional Suitable | Environmental Suitable | Spatial-Functional Suitable | Environmental + Spatial-Functional Unsuitable |
|---|---|---|---|---|
| ORness = 0 | 37,798 | 263,519 | 89,626 | 1572,768 |
| ORness = 0.25 | 170,331 | 420,879 | 166,614 | 1205,888 |
| ORness = 0.5 | 349,870 | 453,225 | 212,734 | 947,882 |
| ORness = 0.75 | 554,041 | 423,467 | 283,380 | 702,823 |
| ORness = 1 | 917,534 | 401,282 | 308,791 | 336,104 |

## 5. Discussion

Determining the current status of UEQ directly represents the quality of life of people in the urban environment [95]. The first and foremost step to provide a suitable solution to improve UEQ is to evaluate the current situation of UEQ. In previous stud-

ies, various models such as GIS-MCDM [26,75,96,97], machine learning [98–100], and regression [100–102] have been used for the spatial evaluation of UEQ. Determining the suitability degree of UEQ is a spatial issue and depends on various spatial factors, including environmental and spatial-functional criteria. Therefore, in this study, a GIS-MCDM model was used to produce the UEQ map.

Previous studies show that the accuracy of GIS-MCDM models depends on the comprehensiveness of the considered effective criteria, the accuracy and consistency of the weights assigned to the criteria, the standardization method of effective criteria values, and the efficiency of aggregation models. The considered effective criteria must enjoy certain conditions, such as comprehensiveness and compatibility. The opinions of experts and the results of previous studies can be used to determine the effective criteria. The accuracy of the weight of the effective criteria has a direct effect on the accuracy of the final output of the decision-making model. Various methods such as AHP [95,103], ANP [104,105], and Best–Worst [106] have so far been used to calculate the weight of criteria. The present study used the AHP method. This method has several advantages, including (i) the simultaneous use of quantitative and qualitative criteria [48], (ii) simple implementation and flexibility [107], and (iii) the ability to check the consistency rate of the results [108]. The consistency rate is one of the outputs of the AHP model, which shows the degree of compatibility among the opinions of different experts in the pairwise comparison of different criteria and criteria weighting. The consistency rate calculated in this study shows the high accuracy and consistency of the determined weights for different effective criteria.

In this study, for the first time, the OWA model was used for modeling and spatial evaluation of UEQ. OWA can model UEQ in a wide range of very optimistic to very pessimistic scenarios. The results of this model can benefit managers and planners with different attitudes. Managers and planners with pessimistic attitudes are usually stricter in setting priorities. Therefore, in this case, the places designated as areas with suitable UEQ conditions are limited since they must have suitable conditions in terms of a large number of effective criteria. As for optimistic managers and planners, if a place has suitable conditions in terms of a small number of effective criteria, it can have suitable UEQ conditions. Therefore, the area of suitable areas is larger in optimistic scenarios than in pessimistic scenarios.

In this study, the criteria affecting UEQ were categorized into two groups: environment and spatial-functional. The results showed that the criteria had different levels of effectiveness concerning urban environmental quality. Due to the high correlation of most of the considered indices with vegetation, it was the most influential index in UEQ. Therefore, to improve other indicators and ultimately UEQ, it is suggested to expand green spaces. However, green roofs are recommended in areas with high density and activities, such as Districts 6 and 11, where there is inadequate space to increase green spaces and create local and regional parks. It can also be suggested to increase tree-lined streets, which increase humidity, create shade, reduce noise pollution, and reduce air pollution. Coniferous trees are recommended as they are green year-round and can better absorb air pollution particles, which is a major problem in urban areas. Still, broad-leaved trees should not be excluded since, in addition to producing more oxygen, these types of trees have higher evaporation and transpiration than conifers and increase the humidity, tranquility, and freshness of the environment.

The dark color of urban surfaces causes strong absorption of solar energy and increases the temperature, thus causing the formation of urban heat islands. Another way to reduce heat, besides increased vegetation, is to replace materials with high reflection instead of dark and impermeable urban surfaces such as sidewalks, streets, and roofs, which have high absorption. By examining the spatial distribution of air pollution, urban heat islands, and ground surface temperature, which have a high correlation, it can be seen that such areas coincide with areas with high building height and building density. Therefore, it is suggested to pay attention to the prevailing wind flow directions when constructing high-rise buildings so as prevent heat retention and air pollution. Improving the status of

infrastructure criteria such as access to parks and medical centers can also be effective in improving the existing UEQ.

The accuracy of benchmark maps used in the OWA model directly affects the accuracy of UEQ modeling; however, access to accurate maps has been one of the limitations of this study. For example, the air pollution benchmark map is prepared based on the interpolation of data recorded in a limited number of ground stations. Also, the height of buildings can be considered an effective criterion in UEQ modeling, which was not used in this study due to a lack of access to these data.

## 6. Conclusions

Today, UEQ is considered a fundamental concept in urban planning, which has attracted the attention of urban planners and managers. This index can help to effectively identify the existing situation, strengths, weaknesses, and possible defects in the urban environment. This study aimed to provide a strategy based on a spatial multi-criteria decision-making system to assess UEQ, that ultimately augments the greater goal of achieving sustainable cities. The results showed that from the environmental point of view, air pollution and distance from faults had the highest and lowest impact in modeling the quality of the urban environment, respectively. Furthermore, in terms of the spatial-functional criteria, population density had the highest, and distance from industrial centers had the least impact in UEQ modeling. The central and eastern districts of the study area had better UEQ conditions than other districts. Districts 2 and 11 had the worst UEQ conditions. By increasing the value of ORness (the degree of optimism), the area of the high and very high UEQ classes increased and the area of the low and very low UEQ classes decreased. In very pessimistic, pessimistic, intermediate, optimistic, and very optimistic scenarios, respectively, 36.64, 49.96, 58.65, 71.37, and 91.24 percent of the study area had suitable UEQ conditions (high and very high class). In the intermediate scenario, respectively, more than 349, 453, 212, and 947 thousand people lived in areas with suitable UEQ conditions in terms of "Environmental + Infrastructure", "Environmental", "Infrastructure", and areas with unfavorable UEQ conditions. The results of this study can be useful for managers and planners to better implement their programs while improving the quality of the urban environment. It is also possible to use the capability of the OWA model to produce UEQ maps under different scenarios to determine the spatial priorities for managers and planners with different conditions, including time and budget. However, determining the weight of the criteria based on a limited number of experts can be associated with uncertainty, so it is recommended to use large group decision-making (LGDM) methods to determine the weight of the criteria in future studies.

**Author Contributions:** Conceptualization, B.M., S.R.T. and F.N.; methodology, B.M, F.N., F.F., A.E., S.A. and M.Z.A.; writing—original draft preparation, B.M., R.A., F.N., F.F., A.E., S.A. and M.Z.A.; writing—review and editing, J.J.A., M.A. and I.M. All authors have read and agreed to the published version of the manuscript.

**Funding:** This research received no external funding.

**Data Availability Statement:** The data used to support the findings of this study are available from the corresponding author upon reasonable request.

**Acknowledgments:** The authors thank anonymous reviewers for their constructive comments and suggestions which helped to improve the manuscript.

**Conflicts of Interest:** The authors declare no conflict of interest.

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
