# Peer review of "A Scenario-Based Spatial Multi-Criteria Decision-Making System for Urban Environment Quality Assessment: Case Study of Tehran"

_land, doi:10.3390/land12091659_

Round 1

Reviewer 1 Report

The paper deals with relevant topic of urban environment quality and its role in the processes of urban planning and development, which makes it appropriate for this journal. Its structure and the methodological approach are correct.

However, the paper requires some minor revisions in order to capitalize on its potential.

I recommend the following:

- Because the research findings refer only to selected districts in Teheran (there is not a generalization of the findings in Conclusion part), please add in the title: Case Study of Teheran.

State the research objective (line 131-135) in a new paragraph. After that, clearly state the research question (they are missing). 

- The Conclusion part requires a wider elaboration. Please, remind the readers of your research question(s) and answer it, comment on the wider relevance of your findings and reflect on implications.

- Bearing in mind what the authors mean by it, the term infrastructure criteria is questionable for me. I would rather recommend the term spatial-functional criteria.

- Please, use the term macro-spatial context instead macro-context (line 72).

- Sources in Figures and Tables are missing. Please, list them. 

Author Response

Our response follows:

Reviewer #1:

Dear reviewer

We appreciate your time and efforts in reviewing the paper and providing constructive comments. We did our best to address your comments and we believe the paper has improved accordingly.  Below, you can see our answers to your comments.

The paper deals with relevant topic of urban environment quality and its role in the processes of urban planning and development, which makes it appropriate for this journal. Its structure and the methodological approach are correct.

However, the paper requires some minor revisions in order to capitalize on its potential.

I recommend the following:

Point 1:  Because the research findings refer only to selected districts in Teheran (there is not a generalization of the findings in Conclusion part), please add in the title: Case Study of Teheran.

Our Answer:

First of all, we are grateful for your positive feedback on our review work. Based on your comment, the title was revised and "case study of Tehran" was added to the title (Please see: Page 1, Title).

Point 2: State the research objective (line 131-135) in a new paragraph. After that, clearly state the research question (they are missing).

Our Answer:

Thank you for pointing this out. Based on your comment, the research objective was transferred to a new paragraph and research questions were added (Please see: Page 3, Line 107-114).

Point 3:  The Conclusion part requires a wider elaboration. Please, remind the readers of your research question(s) and answer it, comment on the wider relevance of your findings and reflect on implications.

Our Answer:

We have revised the conclusion section and elaborated it with the highlighted text. Please see the highlighted text in yellow. (Please see: Page 23, Line 577-581 and 583-597).

Point 4: Bearing in mind what the authors mean by it, the term infrastructure criteria is questionable for me. I would rather recommend the term spatial-functional criteria.

Our Answer:

Thanks for this comment. Changed "Infrastructural criteria" to "Spatial-functional criteria" in all manuscripts.

Point 5: Please, use the term macro-spatial context instead macro-context (line 72).

Our Answer:

Based on the comments of #reviewer 2, the text of this section was rewritten and this phrase was removed from this section.

Point 6: Sources in Figures and Tables are missing. Please, list them.

Our Answer:

We have added the references for all Figures and Tables.

Reviewer 2 Report

This paper proposes a scenario-based spatial multi-criteria decision making system for evaluating the urban environment quality (UEQ). Environmental and infrastructure criteria are selected based on domain knowledge and AHP is used for deciding the weights of them. Then the OWA and GIS are used for evaluating UEQ combing these criteria, and its spatial distribution in study area. Different scenarios including very pessimistic, inter- 32 mediate, optimistic, and very optimistic scenarios are compared and assessed. 

Based on my review, the followings need more consideration and improvement:

1. I would suggest the authors to separate the current Introduction section into 1) Introduction section that introduces the context and significance of this research; and 2) Literature Review section that reviews previous related work, and point out explicitly the difference of this work to previous work, and research gap this one tries to fill. 

2. The first section paragraph in Section 2.3 Methodology reviews different method for UEQ evaluation. I would suggest the authors to move this discussion to the Literature Review section. In addition, this paragraph discusses the capabilities of GIS in spatial decision making process and its limits for multi-criteria decision making. Very relevantly, it is worthy noting that besides the spatial display, GIS does provide some access to spatial decision making that involves network analytics and facility locating problems, with known limits. The authors can refer to:

Church, R. L. (2002). Geographical information systems and location science. Computers & Operations Research29(6), 541-562.

Murray, A. T., Xu, J., Wang, Z., & Church, R. L. (2019). Commercial GIS location analytics: capabilities and performance. International Journal of Geographical Information Science33(5), 1106-1130.

3. Lines 118 - 124 jumps into the discussion on some specific districts in Tehran to demonstrate the importance of UEQ evaluation significance in the study area. While the readers have no context on these specific districts. I would suggest the authors to move the Figure 1 to the introduction section to avoid confusion. 

4. I am curious why low elevation is more preferred for a better UEQ? I am thinking instead of the absolute elevation number, probably the slope or terrain roughness matters more for UEQ?

5. Please make it clear what is the output of the AHP in this particular case, probably with some demonstration in Section 2.3.3. 

6. The description of the OWA method is unclear. In Formula (3), what is the a vector, what is n, what is the output OWA(a_1, a_2, … a_n). Similarly for formula (4), what is OWA_i here. Basically, the authors should define/specify clearly on all notations and parameters used here. 

7. Lines 271-273: “If ? is considered to be the next ? vector and ? represents weights of the second type, the OWA operator works…” the vector a definition and the second type are both confusing here. 

8. I wonder why environment and infrastructure factors are evaluated separately then combined together by the (0.46, 0.56) weights. Why don’t the authors directly consider all criteria (8 environment + 6 infrastructure) together for the final UEQ map? 

9. Figure 14 the legend is not clear. If I understand it correctly, “enforcement + infrastructure” should be “environmental + infrastructural suitable”; “environmental” should be “environmental suitable only”; “infrastructure” should be “infrastructure suitable only”; “unsuitable” should be “both unsuitable”. 

Author Response

Our response follows:

Reviewer #2:

Dear reviewer

We appreciate your time and efforts in reviewing the paper and providing constructive comments. We did our best to address your comments and we believe the paper has improved accordingly.  Below, you can see our answers to your comments.

This paper proposes a scenario-based spatial multi-criteria decision making system for evaluating the urban environment quality (UEQ). Environmental and infrastructure criteria are selected based on domain knowledge and AHP is used for deciding the weights of them. Then the OWA and GIS are used for evaluating UEQ combing these criteria, and its spatial distribution in study area. Different scenarios including very pessimistic, intermediate, optimistic, and very optimistic scenarios are compared and assessed.

Based on my review, the followings need more consideration and improvement:

Point 1: I would suggest the authors to separate the current Introduction section into 1) Introduction section that introduces the context and significance of this research; and 2) Literature Review section that reviews previous related work, and point out explicitly the difference of this work to previous work, and research gap this one tries to fill.

Our Answer:

First of all, we are grateful for your positive feedback on our review work. Based on your comment, the introduction was divided into two parts with the title (1) Introduction (2) Literature Review and new materials were added. (Please see: Page 2-4, Line 49-163).

Point 2: The first section paragraph in Section 2.3 Methodology reviews different method for UEQ evaluation. I would suggest the authors to move this discussion to the Literature Review section. In addition, this paragraph discusses the capabilities of GIS in spatial decision making process and its limits for multi-criteria decision making. Very relevantly, it is worthy noting that besides the spatial display, GIS does provide some access to spatial decision making that involves network analytics and facility locating problems, with known limits. The authors can refer to:

  • Church, R. L. (2002). Geographical information systems and location science. Computers & Operations Research, 29(6), 541-562.
  • Murray, A. T., Xu, J., Wang, Z., & Church, R. L. (2019). Commercial GIS location analytics: capabilities and performance. International Journal of Geographical Information Science, 33(5), 1106-1130.

Our Answer:

Thanks for this comment. Based on your comment, the text in the first section paragraph of Section 2.3 was transferred to the literature review section. Also, information about GIS capabilities was modified based on your feedback (Please see: Page 3, Line 123-140).

Point 3: Lines 118 - 124 jumps into the discussion on some specific districts in Tehran to demonstrate the importance of UEQ evaluation significance in the study area. While the readers have no context on these specific districts. I would suggest the authors to move the Figure 1 to the introduction section to avoid confusion.

Our Answer:

Thank you for pointing this out. Based on your comment, the information in this section was transferred to section 3.1 where the characteristics of the study area are presented (Please see: Page 4, section: 3.1. Study area).

Point 4: I am curious why low elevation is more preferred for a better UEQ? I am thinking instead of the absolute elevation number, probably the slope or terrain roughness matters more for UEQ.

Our Answer:

New information has been added to this section. The elevation criterion was used to determine the comfort level of the living environment. This criterion also affects accessibility. Areas located at a higher elevation have rougher topographical conditions and higher slopes, so the access level in these areas is lower. As a result, the quality of the urban environment decreases with the increase in elevation (Please see: Table 1).

Point 5: Please make it clear what is the output of the AHP in this particular case, probably with some demonstration in Section 2.3.3.

Our Answer:

We have included new information about AHP model output to Section 3.3.3. The final goal of this method is to calculate the weight of criteria and sub-criteria used in a spatial multi-criteria decision-making system (Please see: Page 8 and 9, Line 260-276).

Point 6: The description of the OWA method is unclear. In Formula (3), what is the a vector, what is n, what is the output OWA(a_1, a_2, … a_n). Similarly for formula (4), what is OWA_i here. Basically, the authors should define/specify clearly on all notations and parameters used here.

Our Answer:

Based on your comment, this section has been rewritten. Also, all the variables in the formula were defined precisely (Please see: Page 9, Line 281-295).

Point 7: Lines 271-273: “If ? is considered to be the next ? vector and ? represents weights of the second type, the OWA operator works…” the vector a definition and the second type are both confusing here.

Our Answer:

Based on your comment, this section has been rewritten (Please see: Page 9, Line 281-295).

Point 8: I wonder why environment and infrastructure factors are evaluated separately then combined together by the (0.46, 0.56) weights. Why don’t the authors directly consider all criteria (8 environment + 6 infrastructure) together for the final UEQ map?

Our Answer:

Decision-making and the types of solutions available to improve UEQ conditions are different from each other in terms of environmental and infrastructural. Therefore, presenting results separately for UEQ conditions in terms of environmental and infrastructural can be useful for managers and urban planners in both sectors in terms of implementation and adequate planning. For this reason, in this study, in the first stage, maps of UEQ conditions were prepared separately based on environmental and infrastructural sub-criteria for the study area. Then these maps were combined based on the weight of environmental and infrastructural criteria and the final UEQ map was prepared. Also, by considering the environmental and infrastructure sub-criteria in an integrated way, the rate of inconsistency in the AHP model increases in determining the weight of these sub-criteria. Determining the weight of environmental and infrastructural criteria separately increases the accuracy of determining the weight of these criteria.

Point 9: Figure 14 the legend is not clear. If I understand it correctly, “enforcement + infrastructure” should be “environmental + infrastructural suitable”; “environmental” should be “environmental suitable only”; “infrastructure” should be “infrastructure suitable only”; “unsuitable” should be “both unsuitable”.

Our Answer:

Thank you for your good suggestion. We have regenerated the map based on your input (Please see: Page 20, Figure 13 in the revised version).

Reviewer 3 Report

The study takes into consideration an extremely topical and interesting topic as it intends to present a spatial multi-criteria decision-making system based on scenarios for the evaluation of the UEQ (quality of the urban environment). 5 districts of the city of Tehran were selected as a pilot study. The issue of urban quality is a central theme in sector studies, and I find the connection with climate change that the authors immediately highlighted in the introduction extremely interesting.

In my opinion, the theme and the method used in the study are very original.

The study is clearly organized, contains all components of a scientific article (introduction, materials and methods, results, discussions, and conclusions). Reading the article was interesting but the text needs to be integrated and revised in some points (listed below); in particular, the section relating to the results must be somewhat simplified or otherwise systematized.

The baseline of the study is clearly outlined with a good literature review. The objective of the study is clearly explained and well contextualized with respect to previous research.

In my opinion, the method is the most articulated part of the paper, the section, as described, is quite complete and understandable (perhaps with some clarifications, as indicated below).

The results section is the one that needs the most attention. Compared to the description of the method, this section remains too complex and does not allow for a fluid reading and understanding of the text.

Reading the document was, in my opinion, quite interesting and could interest readers, even from different areas, because the subject lends itself to being explored across the board involving different disciplines.

The topic is topical but, in my opinion, the article needs major revision before publication.

In particular

1. Introduction

In my opinion, the authors have well organized the introduction, highlighted the central themes and contextualized the topics covered with adequate bibliographic references.

From line 93 to line 105: I find the bibliographic references interesting, perhaps I would add further details on the studies that the authors find most interesting (or on the studies that have most influenced the article presented). It would also be important to understand the limits of these works (cited in the text) to better understand the setting of the study in question.

2. Materials and methods

Paragraph 1 (line 137)

In my opinion, the text would be further enriched if the authors dwelt more on the description of the 5 districts selected in the study. I find this first paragraph a little too concise.

Paragraph 3 (line 180)

Lines 181-188: I suggest moving this part to the introduction, where you review the various studies. in the method I would illustrate the original contribution of the authors with the study in question. Figure 2 is very clear and effective.

In general, I consider the section well done. The method is complex, and I think the authors have tried to explain the whole process as much as possible. The selection of criteria is very interesting; I might suggest inserting the measurement method used in the table (not just the reference to the article in the bibliography).

3. Results

In this section we immediately talk about the weights assigned by the experts. How were the weights assigned?

The method used is rather complex and it would be useful to proceed, in the description, according to the scheme drawn in the method.

Lines 358 to 360: must be described in the method

I suggest that authors organize their results more systematically; this would help and facilitate understanding. For example, it would be interesting to describe the results of the 5 districts studied and understand how they relate to each other (as was done in the description of figure n.12).

4. Discussions and 5. Conclusions

The section is well structured, I advise the authors to include any limitations of the study carried out.

The conclusions section is a bit repetitive. I repeat, it would be interesting to understand whether the limits of previous studies have been exceeded, whether the objectives set in the study have been achieved, any limits of the study and any future prospects.

Quality of English Language:

for me the text was clearly understandable and rather fluid.

Minor editing of English language required.

Author Response

Our response follows:

Reviewer #3:

Dear reviewer

We appreciate your time and efforts in reviewing the paper and providing constructive comments. We did our best to address your comments and we believe the paper has improved accordingly.  Below, you can see our answers to your comments.

The study takes into consideration an extremely topical and interesting topic as it intends to present a spatial multi-criteria decision-making system based on scenarios for the evaluation of the UEQ (quality of the urban environment). 5 districts of the city of Tehran were selected as a pilot study. The issue of urban quality is a central theme in sector studies, and I find the connection with climate change that the authors immediately highlighted in the introduction extremely interesting.

In my opinion, the theme and the method used in the study are very original.

The study is clearly organized, contains all components of a scientific article (introduction, materials and methods, results, discussions, and conclusions). Reading the article was interesting but the text needs to be integrated and revised in some points (listed below); in particular, the section relating to the results must be somewhat simplified or otherwise systematized.

The baseline of the study is clearly outlined with a good literature review. The objective of the study is clearly explained and well contextualized with respect to previous research.

In my opinion, the method is the most articulated part of the paper, the section, as described, is quite complete and understandable (perhaps with some clarifications, as indicated below).

The results section is the one that needs the most attention. Compared to the description of the method, this section remains too complex and does not allow for a fluid reading and understanding of the text.

Reading the document was, in my opinion, quite interesting and could interest readers, even from different areas, because the subject lends itself to being explored across the board involving different disciplines.

The topic is topical but, in my opinion, the article needs major revision before publication.

In particular:

1. Introduction

Point 1: In my opinion, the authors have well organized the introduction, highlighted the central themes and contextualized the topics covered with adequate bibliographic references. From line 93 to line 105: I find the bibliographic references interesting, perhaps I would add further details on the studies that the authors find most interesting (or on the studies that have most influenced the article presented). It would also be important to understand the limits of these works (cited in the text) to better understand the setting of the study in question.

Our Answer:

First of all, we are grateful for your positive feedback on our review work. Based on your opinion, the information from the literature review is presented as a separate section in the revised manuscript. In this section, more details of the literature review and the strengths and weaknesses of the previous studies are stated (Please see: Page 3 and 4, Line 114-163)

2. Materials and methods

Point 1: Paragraph 1 (line 137)

In my opinion, the text would be further enriched if the authors dwelt more on the description of the 5 districts selected in the study. I find this first paragraph a little too concise.

Our Answer:

Based on your opinion, new information related to the characteristics of each region was added to this section (Please see: Page 4 and 5, Line 175-183, 188-191, 194-195 and 199-201).

Point 2: Paragraph 3 (line 180)

Lines 181-188: I suggest moving this part to the introduction, where you review the various studies. in the method I would illustrate the original contribution of the authors with the study in question. Figure 2 is very clear and effective.

Our Answer:

The text in the first section paragraph of Section 2.3 was transferred to the literature review section (Please see: Page 3, Line 123-140).

Point 3: In general, I consider the section well done. The method is complex, and I think the authors have tried to explain the whole process as much as possible. The selection of criteria is very interesting; I might suggest inserting the measurement method used in the table (not just the reference to the article in the bibliography).

Our Answer:

Based on your opinion, the method used to prepare the map of each of the criteria is presented in the Table (1). But providing more details about each of the methods will increase the volume of the article and its complexity. As a result, appropriate references have been used for readers to access details (Please see: Table 1).

3. Results

Point 1: In this section we immediately talk about the weights assigned by the experts. How were the weights assigned?

Our Answer:

Based on your opinion, the information related to the weight calculation method was presented in the method section 3.3.3 (Please see: Page 8 and 9, Line 260-276).

Point 2: The method used is rather complex and it would be useful to proceed, in the description, according to the scheme drawn in the method.

Our Answer:

Based on your opinion, the results section corresponding to the general research method was divided into different sub-sections to make it easier for readers to understand the results (Please see: Page 10, Line 317; Page 11, Line 336; Page 13, Line 382; Page 15, Line 411; Page 17, Line 442; Page 21, Line 489).

Point 3: Lines 358 to 360: must be described in the method

Our Answer:

Based on your comment, this information was moved to the method section (Please see: Page 10, Line 309-315).

Point 4: I suggest that authors organize their results more systematically; this would help and facilitate understanding. For example, it would be interesting to describe the results of the 5 districts studied and understand how they relate to each other (as was done in the description of figure n.12).

Our Answer:

Based on your opinion, the situation of different districts was compared with each other in terms of criteria and UEQ (Please see: Page 13, Line 369-374; Page 14, Line 391 and 402-407; Page 15, Line 420-424; Page 16, Line 433-438; Page 17, Line 444-448; Page 19, Line 467-472).

4. Discussions and 5. Conclusions

Point 1: The section is well structured, I advise the authors to include any limitations of the study carried out.

Our Answer:

Based on your comment, the limitations of this study were added to the end of the discussions section (Please see: Page 22, Line 563-568).

Point 2: The conclusions section is a bit repetitive. I repeat, it would be interesting to understand whether the limits of previous studies have been exceeded, whether the objectives set in the study have been achieved, any limits of the study and any future prospects.

Our Answer:

Two questions were designed for this study (Page 3, Line 108-112). In this regard, two research questions were answered in the conclusion section. Also, the application of the results of this study and suggestions for future studies were added at the end of the conclusion section (Please see: Page 23, Line 588-595).

5. Quality of English Language:

Point 1: for me the text was clearly understandable and rather fluid.

Minor editing of English language required.

Our Answer:

Thanks for this nice comment. Based on your comment, the language of the manuscript was reviewed by a native.

Round 2

Reviewer 2 Report

Thanks for responding to my comments. The current version looks good to me. 

Author Response

Reviewer #2:

Dear reviewer

We appreciate your time and efforts in reviewing the paper and providing constructive comments.

Reviewer 3 Report

I am pleased and thank the authors for having considered some suggestions.

First of all, I appreciate the inclusion of the literature review paragraph in the introduction. This paragraph highlights what is on the subject, the limits of the different studies and the starting point of the present research.

Regarding the "Materials and Method" section:

Point 1

I appreciate that the authors have enriched the description a little bit, which could actually be developed even further. No relevant information has been added.

In general, rereading the article I believe that the study is of quality and the changes made by the authors have made the text more fluid.

It remains a complex study, and this is clear in the article which continues to have a rather complex structure.

In my opinion, overall the article may be ready for publication after minor revision.

In general, I don't find excessive problems in using the English language.

Indeed, despite the complexity and articulated structure of the paper, I find that the authors have used a very understandable language and that the quality of the English is quite good.

Author Response

Reviewer #3:

Dear reviewer

We appreciate your time and efforts in reviewing the paper and providing constructive comments. We did our best to address your comments and we believe the paper has improved accordingly.  Below, you can see our answers to your comments.

I am pleased and thank the authors for having considered some suggestions.

First of all, I appreciate the inclusion of the literature review paragraph in the introduction. This paragraph highlights what is on the subject, the limits of the different studies and the starting point of the present research.

Regarding the "Materials and Method" section:

Point 1: I appreciate that the authors have enriched the description a little bit, which could actually be developed even further. No relevant information has been added.

In general, rereading the article I believe that the study is of quality and the changes made by the authors have made the text more fluid.

It remains a complex study, and this is clear in the article which continues to have a rather complex structure.

In my opinion, overall the article may be ready for publication after minor revision.

Our Answer:

According to the presented flowchart, the method of this study includes 5 main steps (Please see: Figure 2). Above the flowchart, these steps are briefly explained (Please see: Line 225-235). Based on your suggestion, the details of the study method according to the flowchart are presented in 5 sub-sections: 3.3.1. Determination of criteria, 3.3.2. Standardization of criteria, 3.3.3. Criteria weight calculation, 3.3.4. OWA method and 3.3.5. Population distribution in UEQ classes (Please see: Line 238, 245, 260, 280 and 312).